# A comparative evaluation of image-to-image translation methods for stain transfer in histopathology

**Igor Zingman** [1]                                          IGOR.ZINGMAN@BOEHRINGER-INGELHEIM.COM
**Sergio Frayle** [2]                                                        SFRAYLE@AISUPERIOR.COM
**Ivan Tankoyeu** [2]                                                    ITANKOYEU@AISUPERIOR.COM
**Segrey Sukhanov** [2]                                                 SSUKHANOV@AISUPERIOR.COM
**Fabian Heinemann** [1]                         FABIAN.HEINEAMANN@BOEHRINGER-INGELHEIM.COM
[1] *Boehringer Ingelheim Pharma GmbH and Co., Biberach an der Riß, Germany*
[2] *AI Superior GmbH, Darmstadt, Germany*

**Editors:** Accepted for publication at MIDL 2023

## Abstract

Image-to-image translation (I2I) methods allow the generation of artificial images that share the content of the original image but have a different style. With the advances in Generative Adversarial Networks (GANs)-based methods, I2I methods enabled the generation of artificial images that are indistinguishable from natural images. Recently, I2I methods were also employed in histopathology for generating artificial images of in silico stained tissues from a different type of staining. We refer to this process as stain transfer. The number of I2I variants is constantly increasing, which makes a well justified choice of the most suitable I2I methods for stain transfer challenging. In our work, we compare twelve stain transfer approaches, three of which are based on traditional and nine on GAN-based image processing methods. The analysis relies on complementary quantitative measures for the quality of image translation, the assessment of the suitability for deep learning-based tissue grading, and the visual evaluation by pathologists. Our study highlights the strengths and weaknesses of the stain transfer approaches, thereby allowing a rational choice of the underlying I2I algorithms. Code, data, and trained models for stain transfer between H&E and Masson's Trichrome staining will be made available at https://github.com/Boehringer-Ingelheim/stain-transfer.

**Keywords:** Image to image translation, stain transfer, virtual staining, histopathology, generative adversarial networks.

## 1. Introduction

Image-to-image translation (I2I) methods (Pang et al., 2022) map images from a source to a target domain, usually preserving semantic information, while changing an image style. With the success of conditional GAN-based image generation technology (Mirza and Osindero, 2014), I2I techniques gained popularity, suggesting a generic approach for tackling diverse computer vision problems such as translation between day and night scenes, colorizing gray-scale images, or reconstruction of an image from its edges (Isola et al., 2017). Especially influential were the methods that were able to learn translation between image domains (e.g. between day and night), given only unpaired examples without scene correspondences between the images from the different domains (Liu et al., 2017; Zhu et al., 2017).

In digital histopathology such methods were employed in a few different scenarios, which we will differentiate into three categories. First, I2I was used for *image normalization* (Zanjani et al., 2018; Kang et al., 2021; Mahapatra et al., 2020; Shaban et al., 2019; BenTaieb and Hamarneh, 2018) and *image augmentation* (Wagner et al., 2021) in order to improve the robustness of image processing systems to variability in staining and image acquisition settings. Image normalization is performed at the inference stage and adapts an image to a reference appearance. On the other hand, image augmentation is performed during the training stage to challenge the system with training examples that vary from the standard appearance (and may appear in real data in the future).

Second, I2I enabled not only the subtle correction of image appearances due to variability in staining and image acquisition, but also the translation between colorization styles due to different types of reagents used for staining (so-called *stain transfer*) (Xu et al., 2019; de Haan et al., 2021; Vasiljevic et al., 2021; Boyd et al., 2022; Lahiani et al., 2019). Since particular types of stains are used for the visualization of specific structures in the tissue (e.g. nuclei, fibrotic tissue etc.), such technology allows to avoid repeated staining with different reagents, when there is a need for the analysis of tissue features that are not emphasized with a single type of staining. Additionally, nowadays histopathological laboratories use systems for the automated evaluation of tissue samples. Since, it is often required to analyze samples stained with the reagents that are different from those used for training, stain transfer algorithms become advantageous (Gadermayr et al., 2018; Brieu et al., 2019). Stain transfer can be considered as an example of domain adaptation (Srinidhi et al., 2021) in digital histopathology, where a system needs to be adapted to process images of tissue samples stained with a different reagent compared to samples used for training the system.

The third and most difficult use of I2I in histopathology is *virtual staining*, when artificial images mimicking stained tissues are generated from unstained tissue samples. The literature targeting this challenging problem is more scarce. Only a few works aimed to generate artificial images mimicking stained tissues from fluorescence (Li et al., 2021; Rivenson et al., 2019) and hyperspectral (Bayramoglu et al., 2017) images were published. In (Li et al., 2020) the authors used paired examples of bright-field images of stained and unstained tissue samples to train a virtual staining system. We expect that the research activity in this emerging domain will yet accelerate in the upcoming years, since the ability to perform virtual staining would have tremendous impact in histopathology (Rivenson et al., 2020).

Substantial effort was already made to quantitatively evaluate the effectiveness of different image normalization and augmentation techniques, using both traditional and GAN-based methods (Tellez et al., 2019; Stacke et al., 2020; Zanjani et al., 2018; Shaban et al., 2019). Unfortunately, stain transfer and virtual staining methods (as categorized above) are lacking a comparative study that quantitatively evaluates a broad range of suitable I2I approaches. Here, we provide such a quantitative comparison of I2I methods for stain transfer. To this end, we compare several GAN-based state-of-the-art (Liu et al., 2017; Zhu et al., 2017; Huang et al., 2018; Lee et al., 2018; Kang et al., 2021; Li et al., 2021; Isola et al., 2017; Shaban et al., 2019; Park et al., 2020) and traditional (Macenko et al., 2009; Vahadane et al., 2015; Reinhard et al., 2001) methods. Particularly, we experiment with the conversion between Masson's Trichrome (MT) and Hematoxylin-Eosin (H&E) staining, see visual examples in Figure 1(a). We evaluate the performance of I2I approaches using complementary quantitative measures (Section 3), the tissue grading errors when integrated

into a computer-aided image analysis pipeline (Section 4), and visual pathologists' analysis (Section 5). Our comparative evaluation outlines the limitations and advantages of different I2I methods, thereby allowing practitioners to properly choose the most suitable one.

## 2. Overview of compared image-to-image translation approaches

We evaluate three stain transfer methods that are based on traditional approaches. Reinhard et al. (2001) presented one of the first approaches for transfering color statistics (**ColorStat**) that was later adopted for stain conversion in histopathology. The authors suggest to transfer the mean and standard deviation of images from the target domain to images from the source domain for each of the three components of LAB color space. Macenko et al. (2009) proposed to handle variability in staining by introducing a color normalization method. It assumes the linear combination of two stains in the optical density (OD) space and uses singular value decomposition to represent OD by the product of the stain and the concentration matrices. The former characterizes the properties of the used stains, while the latter characterizes the strength of the performed staining. Using the stain matrix, estimated from the reference image from the target domain, and the concentration matrix of a source image, the image is normalized to have the appearance of the target domain. Vahadane et al. (2015) build on (Macenko et al., 2009) work, but forces the values of the concentration matrix to be non-negative and sparse, which makes the OD decomposition biologically more plausible. A reference image is usually used to learn the unknown parameters of the traditional methods. Instead of relying on a single reference image, we use all the images in the training dataset from the target domain. For Macenko and Vahadane we averaged the stain matrix and pseudo-maxima of stain concentrations over the training dataset, while for ColorStat we averaged mean and standard deviation.

We also evaluate nine stain transfer methods that are based on Generative Adversarial Networks (GAN) (Goodfellow et al., 2014). We denote GAN's generator as $G$ and discriminator as $D$. $x$ and $y$ are the images from $X$ and $Y$ domains, respectively, between which we want to find a mapping. **Pix2Pix** (Isola et al., 2017) uses a conditional GAN (cGAN) to translate $X \rightarrow Y$, conditioning the discriminator on source images $x \in X$. The discriminator is fed with pairs, either of source and generated $\{x, G(x)\}$ images or of source and target $\{x, y\}$ aligned images. An $L_1$ term is added to the adversarial loss to improve the cGAN performance. One disadvantage of this method is that it requires aligned images $\{x, y\}$ for training. To avoid this limitation, we employ the following trick. We train cGAN to colorize images according to $Y$ using pairs of $y$ and corresponding gray-scale images. To transform $x$ to $y$ we first convert $x$ to gray-scale and then apply the trained cGAN. More specifically, we train cGAN on true pairs $\{AB(y), L(y)\}$, and fake pairs $\{G(L(y)), L(y)\}$, where $L()$ and $AB()$ denote the mapping of an image to the lightness and the two color channels of the LAB color space, and G generates A, B color channels. During inference the trained $G$ generates $G(L(x))$ color channels that are merged with lightness $L(x)$.

The following GAN-based methods, by design, do not require paired images for training. **CycleGAN** (Zhu et al., 2017) proposed training two generators $G : X \rightarrow Y$ and $F : Y \rightarrow X$, enforcing, in addition to two adversarial loss terms, a cycle consistency loss that enforces $F(G(X)) \approx X$ and $G(F(Y)) \approx Y$. **UTOM** (Li et al., 2021) adopts the CycleGAN architecture adding a saliency constraint to the loss function. This constraint forces the

retention of the content mask for source and translated images, which should reduce content distortion during translation. **StainGAN** (Shaban et al., 2019) employed the CycleGAN approach for stain normalization in histological images. In contrast to CycleGAN, a standard ResNet (He et al., 2016) model was used in generator networks. **StainNet** (Kang et al., 2021) leverages StainGAN as a teacher to learn a color pixel-to-pixel mapping with a small convolutional network (using 1x1 convolutions). The $L_1$ loss is used to match the output of StainGAN. **CUT** (Park et al., 2020) proposed patchwise contrastive (PatchNCE) loss as an alternative to the cycle loss. PatchNCE is used to minimize the distance between the feature representations of patches from a source image $x$ and corresponding patches from $G(x)$ relatively to the distance to randomly sampled patches from $x$ at other locations. An additional loss (PatchNCE), applied to images from the target domain, functions as identity loss that prevents the generator from making unnecessary changes.

**UNIT** (Liu et al., 2017) introduces a shared latent space forcing corresponding images from two domains to map to the same latent representation. The architecture consists of two GANs and two Variational Autoencoders (Kingma and Welling, 2014) (VAE) with encoders that generate latent codes and with decoders that are also generators of GANs. The loss function consists of two adversarial losses, two VAE losses, and two VAE-like losses, which implicitly model cycle consistency forcing the distribution of the latent codes of translated and original images to coincide. While UNIT assumes a shared latent space, **MUNIT** (Huang et al., 2018) postulates that only part of the latent space (the content) can be shared across domains whereas the other part (the style) is domain specific. To translate an image to the target domain, its content code is recombined with a random style code in the target style space. The objective comprises an adversarial loss, an $L_1$ reconstruction error in the image space, as well as content and style reconstruction errors in the latent space, all in both directions. Like MUNIT, **DRIT** (Lee et al., 2018) factorizes feature representation space to domain-invariant content and domain-specific attribute (style) spaces. Similarly to MUNIT, the loss function includes an adversarial loss, $L_1$ self-reconstruction (image reconstruction) and $L_1$ latent regression (attribute reconstruction) losses. Cross-cycle consistency forces the twice translated image with swapped attribute features to be close to the source image. An additional content adversarial loss aims at distinguishing the domain membership of content codes. Finally, the KL loss forces attribute codes to obey a normal distribution.

## 3. Comparative evaluation

### 3.1. Evaluation metrics

To evaluate the quality of the generated images we consider two factors: a) How well generated images reproduce the visual appearance of images from the target domain, b) How well a generated image preserves the structure of a source image. To the best of our knowledge, there is no single established metric that covers both factors. Therefore, we selected three metrics that are focused on different aspects: *Structural Similarity Index* (SSIM) (Wang et al., 2004), *first Wasserstein Distance* (WD) (Ramdas et al., 2017), and *Fréchet Inception Distance*(FID) (Heusel et al., 2017). SSIM is a perceptual image quality metric developed to assess the degradation of structural information in processed images. For aligned $x$ and $y$ local neighborhoods (we used $7 \times 7$ size), the index is calculated as

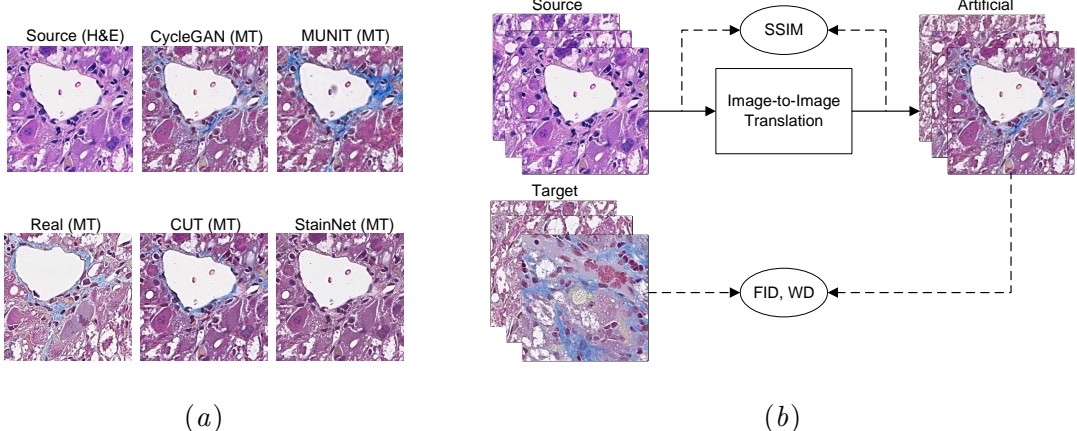

$(a)$ $(b)$

Figure 1: (a) Examples of artificially generated H&E $\rightarrow$ MT images using four I2I methods. Real MT and H&E images were obtained from close slices of tissue. More examples can be found in Appendix A in Figure 2 and Figure 3. (b) Evaluation of I2I: SSIM is applied to pairs of source and artificially generated images converted to gray-scale, while WD and FID are applied to sets of target and artificial images.

follows

$$\text{SSIM}(x, y) = \frac{(2\mu_x\mu_y + c_1)(2\sigma_{xy} + c_2)}{(\mu_x^2 + \mu_y^2 + c_1)(\sigma_x^2 + \sigma_y^2 + c_2)}, \tag{1}$$

where $\mu$, $\sigma^2$, $\sigma_{xy}$ are mean, variance, and covariance of pixel intensities, respectively. $c_1$ and $c_2$ are small constants to avoid instability when the denominator is close to zero. The index was defined as the multiplication of luminance, contrast, and structure factors, which after simplification results in Equation (1). For entire images, SSIM is calculated by averaging along all the local neighborhoods and sometimes termed mean SSIM. Similarly to (Kang et al., 2021), we calculate the mean SSIM between the source and the artificially generated images, both converted to gray-scale, in order to assess structure preservation (see Figure 1$(b)$). We have used the SSIM implementation by Van der Walt et al. (2014).

The WD, also known as Earth-Mover distance, between two one-dimensional discrete distributions $X$ and $Y$ can be computed as follows

$$\text{WD}(X, Y) = \sum_{v \in \mathbb{R}} |C_X(v) - C_Y(v)|, \tag{2}$$

where the $C_X$ and $C_Y$ are cumulative distribution functions. We use WD to measure the discrepancy between color appearances of generated and target images, see Figure 1$(b)$. For this purpose, we average two WDs computed for the two color channels in LAB color space. We have used the WD implementation by Virtanen et al. (2020).

The Fréchet Inception Distance (FID) is a widely adopted metric used to assess the quality of generated images (Parmar et al., 2022). FID compares the distributions of two image sets (e.g. generated and target, see Figure 1$(b)$). Namely, it measures the Fréchet distance (Dowson and Landau, 1982) between the distributions of image deep features generated with the Inception v3 network (Szegedy et al., 2016) pre-trained on the ImageNet (Deng

et al., 2009). Since normal distributions are assumed, FID between distributions $X$ and $Y$ is calculated as follows (Dowson and Landau, 1982)

$$\mathrm{FID}(X,Y) = \|\mu_x - \mu_y\|^2 + tr(\Sigma_x + \Sigma_y - 2(\Sigma_x\Sigma_y)^{\frac{1}{2}}), \tag{3}$$

where $\mu, \Sigma$ are distribution mean and co-variance, and $tr$ is the trace operator. In contrast to the WD described above, FID allows the assessment of not only color but also texture or structure similarity between image sets. We use the Clean-FID implementation (Parmar et al., 2022) in our experiments.

## 3.2. Datasets

To conduct our experiments several datasets of histological images were collected. Whole slide images (WSIs) were acquired with a Zeiss AxioScan scanner (Carl Zeiss, Jena, Germany) with a 20× objective at a resolution of 0.221 µm/pixel from mouse liver tissue samples stained with H&E and MT according to established protocols. The WSI were then subsampled with a factor of 1:2, which resulted in a 0.442 µm/pixel resolution. The *training dataset* consists of around 26000 $256 \times 256$ tiles, extracted from WSIs, for each of the two types of staining. This dataset is used to train I2I methods in both directions.

We collected a disjoint *validation dataset* of around 1300 image tiles for each type of staining. The tiles were extracted from the WSIs used for the training dataset. Therefore, even though there is no overlap between tiles of the training and validation sets, both datasets are coming from the same distribution. We also collected a *test dataset* of around 1300 image tiles for each type of staining extracted from WSIs originating from different histological studies so that the image tiles' distribution is different from the training dataset.

## 3.3. Results

Table 1 summarizes the average translation performance between both directions (H&E → MT and MT → H&E) for the I2I methods. The performance for each direction of translation is separately shown in Appendix C. By calculating the FID measure on the validation dataset with samples distributed similarly to the training data, we conclude that CycleGAN excels over other methods in mimicking the structure and color of target images. CUT and MUNIT performed closely. Our WD measure, which measures the ability to mimic colors (see Section 3.1), shows a similar ranking of the best performing methods. The StainGAN method shows close but slightly worse results than CycleGAN, the design of which was borrowed by StainGAN. UTOM, which also adopted the CycleGAN architecture but introduced an additional constraint to reduce the distortion of image content, did not show (based on SSIM) the expected improvement. The SSIM measure shows that Pix2Pix, StainNet, and ColorStat introduce the lowest level of distortion. However, these methods are essentially worse in generating the desired color and texture. StainNet, with $1 \times 1$ filters, as well as all three traditional methods frequently fail to properly generate color patterns, because they are based on pixel-to-pixel mappings which do not take into account a local neighborhood. Specifically, they cannot reproduce blue patterns of connective tissue for H&E → MT translation, see Figure $1(a)$, and Figure 2, Figure 3 in Appendix A. As measured by SSIM, the lowest distortion among traditional methods, introduced by

Table 1: Evaluation of I2I methods with FID (texture & color similarity to target domain), WD (color similarity to target domain), and SSIM with standard errors (structure preservation) for the Validation, and Test sets. The methods are ordered according to FID for the validation set. Best results are in bold. WD has a factor $10^{-4}$.

| Model | Validation | | | Test | | |
|---|---|---|---|---|---|---|
| | FID↓ | WD↓ | SSIM↑ | FID↓ | WD↓ | SSIM↑ |
| CycleGAN | **16.33** | **1.46** | 0.951 ± 0.001 | **25.18** | **5.04** | 0.934 ± 0.001 |
| CUT | 17.10 | 1.60 | 0.914 ± 0.001 | 29.75 | 6.30 | 0.901 ± 0.001 |
| MUNIT | 19.20 | 1.61 | 0.871 ± 0.001 | 29.36 | 6.15 | 0.842 ± 0.001 |
| StainGAN | 19.59 | 3.27 | 0.952 ± 0.000 | 26.64 | 6.97 | 0.926 ± 0.001 |
| UNIT | 20.23 | 2.54 | 0.940 ± 0.001 | 36.78 | 7.40 | 0.918 ± 0.001 |
| UTOM | 20.64 | 2.32 | 0.952 ± 0.000 | 32.79 | 7.06 | 0.951 ± 0.000 |
| DRIT | 22.83 | 2.06 | 0.915 ± 0.001 | 33.62 | 5.44 | 0.892 ± 0.001 |
| Pix2Pix | 48.47 | 8.42 | **0.998** ± 0.000 | 49.71 | 5.34 | **0.997** ± 0.000 |
| StainNet | 50.49 | 11.41 | 0.972 ± 0.000 | 47.81 | 12.33 | 0.967 ± 0.000 |
| ColorStat | 62.13 | 9.60 | 0.974 ± 0.001 | 58.42 | 6.96 | 0.939 ± 0.001 |
| Macenko | 70.39 | 12.90 | 0.926 ± 0.001 | 53.27 | 11.83 | 0.910 ± 0.001 |
| Vahadane | 76.55 | 15.14 | 0.911 ± 0.001 | 59.94 | 14.71 | 0.885 ± 0.001 |

ColorStat, is similar to the one introduced by CycleGAN and its derivatives UTOM and StainGAN. This reinforces the suitability of CycleGAN for stain transfer.

The results on the test set again demonstrate that CycleGAN achieves the best performance. However, FID and WD show worse values since the generated colors resemble images of the training/validation sets rather than the images of the test set. For the same reason, WD, which exclusively measures color similarity, showed a different ranking of the methods. MUNIT and DRIT performed worse than the principally different CycleGAN. However, they may allow the adaptation to the color distribution of a target domain without the need to retrain an I2I network.

## 4. Robustness of computer-aided grading of tissue to artificial images

A growing number of tasks in quantitative tissue analysis, such as disease grading, are being performed with the aid of machine learning systems. We investigated whether artificial images can be utilized by such systems. This would allow to apply such systems when the stain used for training is not readily available. We use a deep-learning-based system (Heinemann et al., 2019) that was trained on MT stained tissues to replace pathologist grading of non-alcoholic fatty liver disease, which includes a quantification of liver inflammation. The condition, if exists, is typically spread homogeneously over tissue. The system was fed with artificially created H&E → MT images. It analyzes $300 \times 300$ tiles and assigns inflammation scores in the $[0, 2]$ range, which are then averaged for the whole tissue sample. Our study is based on 77 rodent liver tissue sections (used also for the sampling of 1300 tiles for the test dataset, see Section 3.2). In Table 2 we report the Mean Square Error (MSE) and Mean Absolute Error (MAE) between the inflammation scores generated by the system fed with real and artificial MT stained tissue images. We did not perform an analysis of Pix2Pix and UTOM, because the generator implementations did not allow to process $300 \times 300$ image sizes. When the system was fed with (real) H&E tiles (instead of required MT tiles), we obtained a MSE = 0.031, MAE = 0.143. Therefore, the methods performing worse do not

Table 2: MSE, first row, and MAE, second row, of inflammation score [0, 2] when the system was fed with real versus generated MT images. MSE, MAE, and the standard error have a factor $10^{-2}$.

| CycleGAN | UNIT | StainGAN | DRIT | MUNIT | CUT | StainNet | ColorStat | Macenko | Vahadane |
|---|---|---|---|---|---|---|---|---|---|
| $1.1 \pm 0.2$ | $1.2 \pm 0.2$ | $1.3 \pm 0.2$ | $1.7 \pm 0.2$ | $2.4 \pm 0.7$ | $3.3 \pm 0.6$ | $3.7 \pm 0.3$ | $5.3 \pm 1.5$ | $9.7 \pm 0.7$ | $10.3 \pm 0.7$ |
| $8.6 \pm 0.7$ | $8.9 \pm 0.8$ | $8.9 \pm 0.8$ | $11.3 \pm 0.8$ | $11.2 \pm 1.2$ | $13.3 \pm 1.4$ | $17.2 \pm 1.0$ | $14.5 \pm 2.1$ | $28.6 \pm 1.4$ | $29.6 \pm 1.4$ |

provide a benefit, compared to the frequently available H&E stain. In alignment with the results from Section 3.3, CycleGAN, derived from it StainGAN, and UNIT showed strong performance with MSE and MAE close to 0 (the theoretical optimum), while the traditional methods performed poorly. The success of CycleGAN, StainGAN, and UNIT can be attributed to high SSIM measures and simultaneously low values of FID, see Table 1.

## 5. Pathologist assessment

To understand how well human experts can distinguish between artificial and real images of stained tissue, we performed an assessment by pathologists. We asked two pathologists (P1 with 6 years of experience, P2 with 17 years of experience) to identify the 200 artificial images in a mixture of 200 real and 200 artificial images generated by CycleGAN and MUNIT. The images were sampled from the validation set. As shown in Table 3, it was challenging for both pathologists to identify the artificial images. The results for CycleGAN were close to random guessing, whereas for MUNIT it was possible to identify the artificial images in some cases.

Table 3: Pathologists test: Accuracy of the identification of 200 artificial images in a mix of 200 artificial and 200 real images.

| | MT→H&E | | H&E→MT | |
|---|---|---|---|---|
| | CycleGAN | MUNIT | CycleGAN | MUNIT |
| P1 | 0.515 | 0.545 | 0.495 | 0.535 |
| P2 | 0.53 | 0.57 | 0.53 | 0.66 |

## 6. Conclusion

In our study, we evaluated three traditional and nine GAN-based I2I methods for stain transfer in histopathology. The analysis was based on three quantitative measures that assess the quality of color and texture translation, as well as the distortion of the image content. We additionally evaluated the performance of a deep learning grading system that was fed with artificially stained tissue images. Furthermore, we conducted experiments where expert pathologists were asked to distinguish real from artificial images.

The results have shown that CycleGAN provides the highest quality of stain transfer and introduces similar or lower distortions than traditional pixel-to-pixel methods. On the contrary, pixel-to-pixel methods, i.e., StainNet and the traditional methods, are hardly suitable for stain transfer. Moreover, all compared approaches derived from CycleGAN did not show advantages over the original version.

Our study inspires the use of stain transfer methods for both pathologist visual evaluation and computer-aided assessment when a type of staining is missing. Trained models, inference code, and data will accompany this paper. We encourage stain transfer researchers to use our framework for the evaluation of stain transfer methods not included in our study. For example, a potential of emerging diffusion-based methods for stain transfer has not yet been shown. We plan to further experiment with stain transfer going from tiles to WSIs and transferring different types of staining. This would allow pathologists to draw their conclusions faster by multiplexing between several types of staining.

## Acknowledgments

We would like to thank Dr. Birgit Stierstorfer (Non Clinical Drug Safety, Boehringer Ingelheim, Biberach, Germany) and Dr. Charlotte Lempp (Drug Discovery Sciences, Boehringer Ingelheim, Biberach, Germany) for their help with the visual evaluation of the artificially generated images and Dr. Martin Lenter (Drug Discovery Sciences, Boehringer Ingelheim, Biberach, Germany) for helpful discussions and project support. We would also like to thank Dr. Lina Humbeck (Medicinal Chemistry, Boehringer Ingelheim, Biberach, Germany) for her comments, which helped to improve the quality of the paper.

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

# Appendix A. Examples of generated Masson's Trichrome images

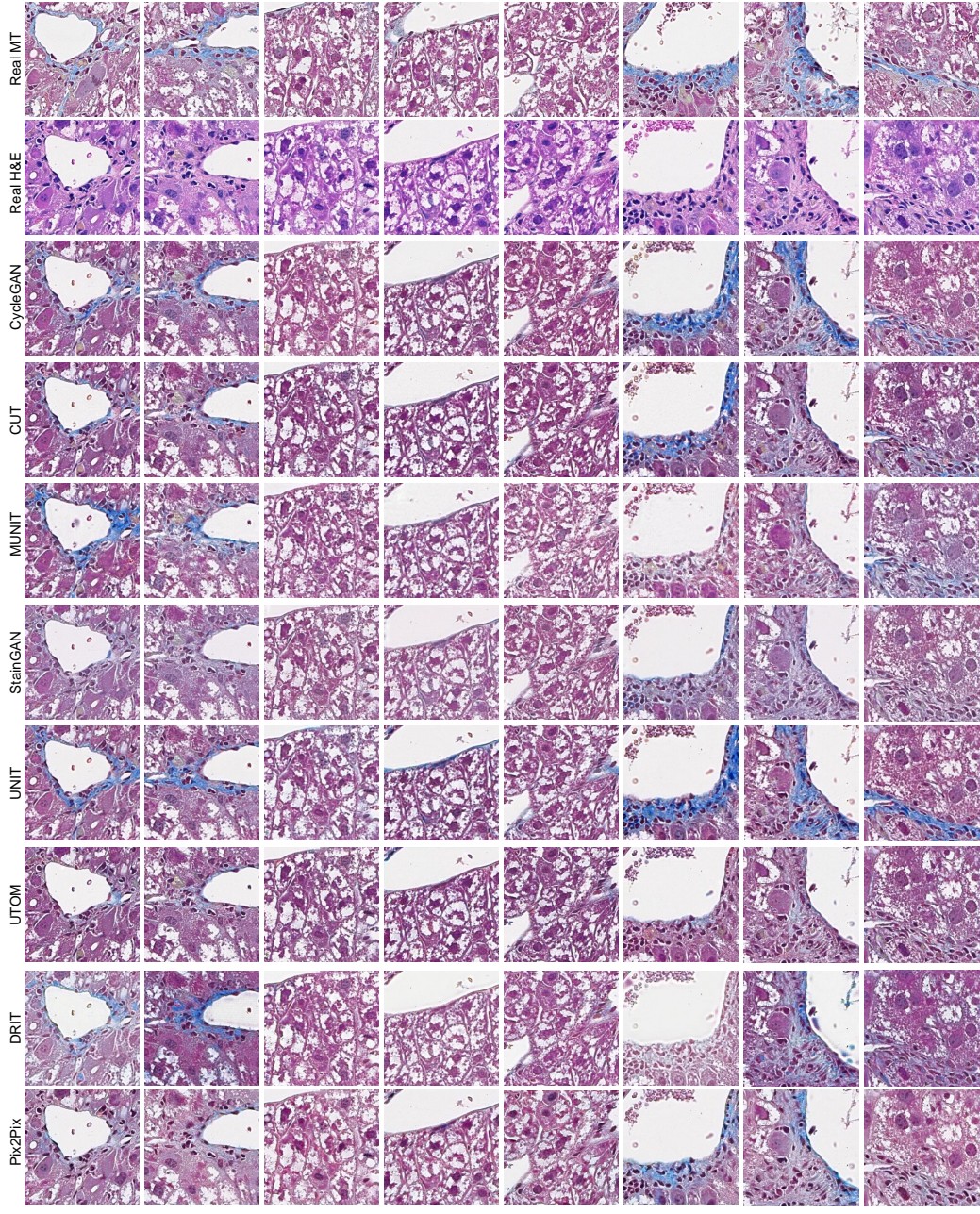

Figure 2: Examples of artificially generated H&E → MT images using GAN-based I2I methods. Real MT and H&E images in the first two rows were obtained from close slices of the tissue.

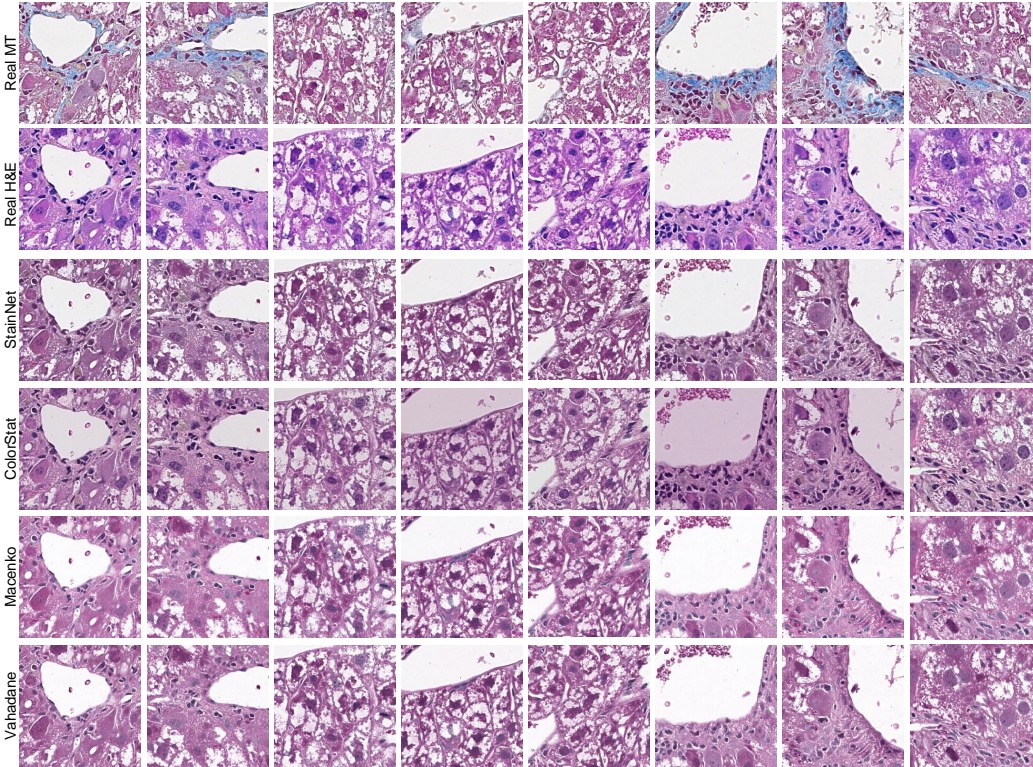

Figure 3: Examples of artificially generated H&E → MT images using pixel-to-pixel I2I methods. Real MT and H&E images in the first two rows were obtained from close slices of the tissue.

## Appendix B. Training details

For all methods, we used the original code available from corresponding online repositories. The training guidelines provided by the authors were followed, except for the methods mentioned below. For MUNIT we removed the spectral weight normalization in the generator to avoid the erroneous inversion of the bright and dark parts of the tissue. For UTOM we adapted the saliency thresholds to background and foreground intensities.

A few training hyperparameters (including the number of training epochs) for some I2I methods deviate from the recommended ones. They were adjusted based on the validation set, which we used to ensure acceptable quality of generated histological images relying on visual inspection and the FID score. The resulting number of epochs, as well as training and inference times are outlined in Table 4. Other parameters will be outlined in the code repository accompanying this paper. For our experiments we used a machine with NVIDIA T4 GPU, 16 GB RAM (AWS 4DN Extra Large instance).

Table 4: The number of training epochs, time per epoch, total training time, the number of network parameters and the inference times.

| Model | Training | | | Inference | | |
|---|---|---|---|---|---|---|
| | Epochs | Epoch (hours) | Time (days) | Params | GPU (s) | CPU (s) |
| CUT | 30 | 1.63 | 2.04 | 11.38 M | 0.034 | 0.876 |
| ColorStat | 1 | 0.059 | 0.0024 | 6 | | 0.009 |
| CycleGAN | 40 | 3.34 | 5.56 | 11.38 M | 0.027 | 0.571 |
| DRIT | 300 | 0.27 | 3.5 | 21.27 M | 0.035 | 0.597 |
| Macenko | 1 | 1.27 | 0.0528 | 8 | | 0.317 |
| MUNIT | 46 | 2.66 | 5.24 | 30.26 M | 0.045 | 0.833 |
| Pix2Pix | 30 | 0.18 | 0.23 | 54.41 M | 0.010 | 0.115 |
| StainGAN | 40 | 4 | 6.67 | 11.38 M | 0.028 | 0.560 |
| StainNet | 300 | 0.029 | 0.36 | 1.28 K | 0.002 | 0.009 |
| UNIT | 115 | 3.12 | 14.95 | 12.56 M | 0.030 | 0.474 |
| UTOM | 200 | 0.57 | 4.71 | 54.41 M | 0.007 | 0.113 |
| Vahadane | 1 | 7.95 | 0.3313 | 8 | | 2.276 |

## Appendix C. Results for each direction of stain transfer

Table 5 summarizes the translation performance of the I2I methods for each direction of translation (H&E → MT and MT → H&E), separately. Here, we also show the metrics for the training dataset, which may facilitate training on other histopathology datasets. Note that substantially lower FID values for the training set are due to the high sensitivity of FID to image set sizes (Binkowski et al., 2018).

Table 5: Evaluation of I2I methods with FID (texture and color similarity to target domain), WD (color similarity to target domain) and SSIM with standard errors (structure preservation of source domain) for the Train, Validation and Test sets. The first row corresponds to H&E → MT, the second to MT → H&E translation directions. The methods are ordered according to FID for the validation set. Best results are highlighted in bold. WD has a factor $10^{-4}$.

| Model | Train | | | Validation | | | Test | | |
|---|---|---|---|---|---|---|---|---|---|
| | FID↓ | WD↓ | SSIM↑ | FID↓ | WD↓ | SSIM↑ | FID↓ | WD↓ | SSIM↑ |
| CycleGAN | **2.65** | **1.29** | 0.950± 0.0 | **16.76** | **1.37** | 0.950 ± 0.001 | **28.47** | 7.41 | 0.947 ± 0.001 |
| | **2.83** | **1.30** | 0.953± 0.0 | **15.89** | 1.55 | 0.953 ± 0.000 | **21.89** | **2.68** | 0.921 ± 0.001 |
| CUT | 3.87 | 1.76 | 0.917± 0.0 | 17.64 | 1.81 | 0.916 ± 0.001 | 31.73 | 7.42 | 0.921 ± 0.001 |
| | 3.57 | 1.46 | 0.910± 0.0 | 16.56 | 1.38 | 0.912 ± 0.001 | 27.76 | 5.18 | 0.882 ± 0.001 |
| MUNIT | 6.28 | 1.53 | 0.872± 0.0 | 19.23 | 1.67 | 0.871 ± 0.001 | 31.69 | 6.76 | 0.875 ± 0.001 |
| | 6.55 | 1.35 | 0.869± 0.0 | 19.18 | 1.56 | 0.871 ± 0.001 | 27.03 | 5.54 | 0.808 ± 0.002 |
| StainGAN | 5.92 | 4.42 | 0.952± 0.0 | 19.65 | 4.72 | 0.953 ± 0.000 | 30.40 | 7.08 | 0.955 ± 0.000 |
| | 6.38 | 1.71 | 0.951± 0.0 | 19.53 | 1.83 | 0.951 ± 0.001 | 22.88 | 6.86 | 0.898 ± 0.001 |
| UNIT | 6.53 | 3.80 | 0.951± 0.0 | 20.39 | 3.84 | 0.951 ± 0.000 | 42.42 | 7.13 | 0.957 ± 0.000 |
| | 7.53 | 1.31 | 0.929± 0.0 | 20.07 | **1.23** | 0.929 ± 0.001 | 31.14 | 7.67 | 0.880 ± 0.002 |
| UTOM | 7.43 | 2.59 | 0.950± 0.0 | 20.81 | 3.02 | 0.950 ± 0.000 | 39.88 | 6.35 | 0.959 ± 0.000 |
| | 7.50 | 1.47 | 0.955± 0.0 | 20.48 | 1.62 | 0.955 ± 0.000 | 25.70 | 7.77 | 0.944 ± 0.000 |
| DRIT | 12.88 | 2.21 | 0.910± 0.0 | 25.47 | 1.93 | 0.912 ± 0.001 | 44.06 | 8.09 | 0.919 ± 0.001 |
| | 6.77 | 1.71 | 0.919± 0.0 | 20.19 | 2.18 | 0.919 ± 0.001 | 23.18 | 2.78 | 0.865 ± 0.002 |
| Pix2Pix | 34.44 | 8.30 | **0.998**± 0.0 | 49.82 | 8.62 | **0.998** ± 0.000 | 50.79 | **4.78** | **0.998** ± 0.000 |
| | 33.69 | 7.96 | **0.998**± 0.0 | 47.11 | 8.21 | **0.998** ± 0.000 | 48.63 | 5.89 | **0.997** ± 0.000 |
| StainNet | 31.61 | 13.27 | 0.971± 0.0 | 45.83 | 13.34 | 0.971 ± 0.000 | 37.88 | 11.22 | 0.973 ± 0.000 |
| | 42.56 | 9.56 | 0.973± 0.0 | 55.15 | 9.49 | 0.973 ± 0.000 | 57.74 | 13.44 | 0.960 ± 0.000 |
| ColorStat | 45.30 | 12.00 | 0.977± 0.0 | 59.66 | 12.55 | 0.977 ± 0.001 | 49.82 | 10.29 | 0.979 ± 0.001 |
| | 50.67 | 6.62 | 0.971± 0.0 | 64.60 | 6.65 | 0.971 ± 0.001 | 67.02 | 3.64 | 0.899 ± 0.002 |
| Macenko | 46.65 | 21.41 | 0.918± 0.0 | 60.45 | 21.24 | 0.918 ± 0.001 | 34.70 | 19.42 | 0.910 ± 0.001 |
| | 67.14 | 4.33 | 0.934± 0.0 | 80.33 | 4.56 | 0.934 ± 0.001 | 71.83 | 4.25 | 0.910 ± 0.002 |
| Vahadane | 51.87 | 24.90 | 0.903± 0.0 | 65.98 | 24.73 | 0.902 ± 0.001 | 38.70 | 25.32 | 0.885 ± 0.001 |
| | 71.82 | 5.54 | 0.915± 0.0 | 87.11 | 5.55 | 0.919 ± 0.001 | 81.18 | 4.10 | 0.885 ± 0.002 |

