# OpenReview forum: "A comparative evaluation of image-to-image translation methods for stain transfer in histopathology"
_MIDL.io/2023/Conference — MIDL 2023 Poster_

### Official Review · Reviewer_Unus · 2023-02-03

**Confidence:** 3
**Preliminary Rating:** 3
**Recommendation:** Poster

**Summary:**

This paper compares three traditional and nine Generative Adversarial Network-based H&E to MT stain transfer approaches whose quality performance has been quantitatively assessed. First, the authors evaluate the output of the models with Structural Similarity Index (SSIM), Fréchet Inception Distance (FID), and first Wasserstein Distance (WD) by comparing it with its corresponding reference image for the whole dataset. Secondly, the reference MT stained tissue images and synthetically-produced tissue images are fed into a stain-level classifier, and Mean Squared Error (MSE) is calculated by using the outputs of the classifier. Finally, a reader's study is performed on the outputs of CycleGAN and MUNIT to investigate whether the synthetic images are indistinguishable from the authentic ones.

**Strengths:**

1. The paper is generally well-written
2. Code to be shared with the community
3. Evaluated on a downstream stain tissue classification task
4. A reader's study has been performed to validate that the produced images by the CycleGAN model are as satisfactory as the authentic images

**Weaknesses:**

1. Only GAN-based approaches have been compared.
2. Assumptions about some of the experiments have not been stated.
3. Some additional metrics are required to analyze the classification performance.


**Deanonymize Review:**

no

**Detailed Comments:**

1. Can you explicitly state the contributions of your research paper?
2. It is not surprising that CycleGAN and Pix2Pix tend to perform better than the other methods in terms of image quality, and there are existing research papers already demonstrating that (de Haan et. al., 2021) and (Salehi et. al., 2021). It would be more interesting to see whether it is possible to improve these results with recent generative models for I2I, such as Latent Diffusion Models (Rombach et. al., 2021).
3. Sufficient selection of metrics in terms of measuring quality, but cannot say the same thing for the classification tests.
4. I couldn’t understand the trick you did on Pix2Pix and how using that trick helped you to overcome the aligned image prerequisite. Can you elaborate on that a bit further?
5. Are you planning to experiment on other stain types, such as periodic acid-Schiff and Jones silver stain?
6. Do you have the ground-truth inflammation scores for the 1300 image tiles used for testing the artificially stained tissues? If so, can you provide the changes in the classification accuracy/precision/recall/$F_1$-score? If not, I assume that you calculate the MSE of the model outputs between authentic and synthetic stained tissues, and I would like to ask how you can be sure that Heinemann et al.'s system is working effectively for your testing split.
7. Can you mention the years of expertise of the pathologists in Section 5?
8. Are you also planning to open-source the dataset alongside the source code?
9. What is the reason for you to select MUNIT and CycleGAN for the pathologist assessment?

- Kevin de Haan, Yijie Zhang, Jonathan E. Zuckerman, Tairan Liu, Anthony E. Sisk, Miguel F. P. Diaz, Kuang-Yu Jen, Alexander Nobori, Sofia Liou, Sarah Zhang, Rana Riahi, Yair Rivenson, W. Dean Wallace, Aydogan Ozcan. “Deep learning-based transformation of H&E stained tissues into special stains”. Nature Communications 12 (2021).
- Pegah Salehi, Abdolah Chalechale. “Pix2Pix-based Stain-to-Stain Translation: A Solution for Robust Stain Normalization in Histopathology Images Analysis”. MVIP 2020.
- Robin Rombach, Andreas Blattmann, Dominik Lorenz, Patrick Esser, Björn Ommer . “High-Resolution Image Synthesis with Latent Diffusion Models”. CVPR 2022.

**Reason for the change in rating**: Improvement in writing, making satisfactory explanations and the share of code. However, I am still feeling unconvinced by how the use of only GAN-based generative models makes this study interesting, considering that the development of generative models is an active research area.

**Paper Type:**

validation/application paper

**Questions To Address In The Rebuttal:**

Please consider reviewing the points in the detailed comments during the rebuttal period. Especially, I would be looking for the results of latent diffusion models and classification measures other than the mean squared error.

---

### Official Review · Reviewer_QQ4A · 2023-02-04

**Confidence:** 4
**Preliminary Rating:** 3
**Recommendation:** Poster

**Summary:**

The paper categorizes histopathology image2image translation applications into three families: 1) image normalization/augmentation, 2) stain transfer, and 3) virtual stain transfer. After claiming that benchmarking has only been done for 1), they proceed to assess and compare lots of image2image techniquers in this context, for a particular dataset and two stains.

**Strengths:**

* The list of I2I techniques that have been implemented and assessed is impressively long.
* Two pathologists were involved in human evaluation
* A general, very popular, and well-accepted technique (cycle-gan) perform better than specialized approaches, which I find consistent with a stream of literature showing that many resarch advances are brittle and probably overfitted to certain datasets.
* The paper may have practical interest for practitioners in the area

**Weaknesses:**

The main problem I find is in the motivation to carry out this analysis. There is a distinction into three families of problems, but I am not sure I agree what the way that distinction has been laid out. Note that the entire paper is based on that taxonomy, since the authors state that no one has benchmarked two of the subcategories, and they are doing so. Please see below for details about this point.

**Deanonymize Review:**

yes

**Detailed Comments:**

This benchmarking paper is based upon a division of I2I applications in histopathology into three families, as explained in the intro. According to the authors, we have:

1) Image Normalization & Image Augmentation: normalization is used in inference time, augmentation during training. Note that for illustrating normalization, they cite the paper (BenTaieb and Hamarneh, 2018), with the title of "Adversarial stain transfer for histopathology image analysis".
2) Stain Transfer, which they describe as methods to re-stain a slide so that it enables quantitative analysis of different structures and tissues, without having to re-acquire.
3) Virtual staining, which is the same as 2) but starting from unstained samples.

If I understand correctly, as opposed to 1), when we do 2) or 3) our goal is not to generate new training data, or adapt test samples at inference time, but rather enable pathologists to read the image in different stains, is this right?

If my understanding is correct, then:
* Shouldn't we have an evaluation in which we try to understand if re-stained (translated) images actually allow to distinguish structures that were undistinguishable with the "old staining"? Running some quantitative analysis like as a histopathologist would do, instead of looking at SSIM, WD, and FID?
* Why does the paper have a section on computer-aided diagnosis? In section 4 the authors adapt in test time test samples and improve the diagnostic, but doesn't this fit in the Image Normalization category inside 1) above?

----

I have some other minor commments, please see below:

- I think it would be a good idea for the sake of readability to add in between parenthesis and maybe with a smaller fontsize, after each number in Table 1 the corresponding rank 1st-12th.
- I find section 4 to be a bit poorly written; what is the system here, maybe give a short description? Can we really average all small tiles' inflammation scores into a single slide-level score in the [0,2] range, and this is meaningful? I mean, don't the score 0 tiles overwhelm the other ones in the averaging? Also, is the MSE computed at the tile or at the slide level? Finally, why couldn't Pix2Pix and UTOM handle 300x300 image sizes? it does not sound like a lot to deal with, right?
-"first Wasserstein Distance" -- remove "first"?
- The reference to (Mirza&Osindero 2014) in the intro to illustrate "recent success" does not feel too recent, it's been nine years already, lots of things have happened...

**Paper Type:**

validation/application paper

**Questions To Address In The Rebuttal:**

I am happy to review my score if the authors can clarify to me the above questions about the motivation and suitability of their benchmarking.
It seems that openreview requires 200 characters here, but what I would like to be addressed has been detailed above already.

---

### Official Review · Reviewer_rkaw · 2023-02-05

**Confidence:** 5
**Preliminary Rating:** 4
**Recommendation:** Poster

**Summary:**

This paper provides a comprehensive evaluation of 12 different stain transfer methods for pixel-to-pixel translation in histopathology. The evaluation is based on a large dataset of over 1000 images. Both quantitative and pathologist assessments are included in the paper for a comprehensive comparison.

**Strengths:**

1. The paper is very well-written. The structure is clear and easy to follow.
2. Not only quantitative evaluation on stain transfer is performed, but also the effect of stain transfer on downstream classification tasks as well as pathologist assessment are presented. This provides a comprehensive evaluation on the 12 stain transfer methods.

**Weaknesses:**

Stain transfer is an important preprocessing step in computational histopathology that helps to address the domain gap. It is crucial to assess its impact on subsequent tasks. This study aims to compare different stain transfer algorithms, therefore, it would be beneficial to include additional comparisons with downstream tasks such as histopathology image diagnosis. Furthermore, as shown in Table 2, many stain transfer algorithms result in a decline in performance. It is suggested to include an analysis of why these algorithms are not effective. In other words, the factors that contribute to the success of StainGAN and UNIT should be discussed.

**Deanonymize Review:**

no

**Detailed Comments:**

Please refer to the comments in the Weakness section.

**Paper Type:**

validation/application paper

**Questions To Address In The Rebuttal:**

Stain transfer is an important preprocessing step in computational histopathology that helps to address the domain gap. It is crucial to assess its impact on subsequent tasks. This study aims to compare different stain transfer algorithms, therefore, it would be beneficial to include additional comparisons with downstream tasks such as histopathology image diagnosis. Furthermore, as shown in Table 2, many stain transfer algorithms result in a decline in performance. It is suggested to include an analysis of why these algorithms are not effective. In other words, the factors that contribute to the success of StainGAN and UNIT should be discussed.

---

### Official Review · Reviewer_g67C · 2023-02-07

**Confidence:** 4
**Preliminary Rating:** 5
**Recommendation:** Poster

**Summary:**

This paper performs a thourough benchmark of different methods for image-to-image translation, in the context of histopathology staining (back and forth between MT and H&E (full disclaimer: I have no idea what those are)).

The authors do a (I think) extensive coverage of the litterature over time (spanning 20 years) and implement 12 methods in a common framework (to be released publicly). The evaluation is made on a dedicated datasets, and completed with an evaluation of performances in a typical use-case, plus another evaluation by clinicians (to spot the "fake" images).

**Strengths:**

- Extensive coverage of the existing methods, including non-deep-learning/GAN based methods
- Dedicated and consistent dataset collected for the purpose of the study
- Thorough evaluation on multiple metrics

**Weaknesses:**

The experiments could benefit from multiple runs, and maybe some pairwise statistical test, to see if the difference between methods are statistically significant.

---
Not a weakness of the paper per se (it is a well executed benchmark), but with the overall field of image-to-image translation, outside of artistic applications: I do not see the point. See my detailed comment in the "questions to address in rebuttal". This will _not_ affect my final rating, but I genuinely wish to engage in this discussion, as so far I have always been quietly dubious about such applications.

**Deanonymize Review:**

no

**Detailed Comments:**

Misc:
- Tables could benefit from standard deviation information
- Maybe indicating if the difference between CycleGAN/CUT, CUT/MUNIT, MUNIT/StainGAN (and so on in Table 1) are statistically significant, would be good.

**Paper Type:**

validation/application paper

**Questions To Address In The Rebuttal:**

To be blunt, I simply do not get the point of image-to-image translation, in a context where we _measure_ something.

For instance, going from my stickman drawings, to a Rembrandt style painting (through stablediffusion and img2img methods) is really interesting, but ultimately everything is "made-up" so accuracy does not matter apart from artistic considerations (I am a very bad drawer).

Here, as far as I am aware, we are trying to visualize, measure, and interpret something real. The stains, from my understanding, are required to highlight what we want to look at, and different staining agents will highlight different parts of the slice. (Another disclaimer: last time I stained something and looked at it in the microscope, I was in high-school.) The paper seems to state the same:

> Since particular types of stains are used for the visualization of specific structures in the tissue (e.g. nuclei, fibrotic tissue etc.),

So while we can make things "look like" another type of stains, the underlying information simply isn't there. (Otherwise we would not bother having different types of stains in the first place?) (I also realize this might motivate the use of cameras that also go into the infrared and ultraviolet spectrum, coupled with fake colors for human interpretation. Is that a thing?) I would argue that "learnable" methods are worse, because they can end-up "injecting" information that isn't there in the first place, spouting its own bias from the training set.

To give a very concrete example, outside of medical applications, last year most of us saw in awe the images produced by the James Webb Space Telescope. It is worth noting that it is not simply a new Hubble with a higher resolution and better optics. The main difference is that the two telescopes do not cover the same range of the light spectrum (JWST being limited to non-visible infrared, from 0.6 to 28 microns, while Hubble in infrared mode could do only from 0.8 to 2.5 microns [1]). Therefore, it can measure things that Hubble could not, and vice versa. Both are complementary (as Hubble is great for ultraviolets).
This is nicely illustrated, I think, with such a comparison image: https://www.nasa.gov/image-feature/goddard/2022/nasa-s-webb-captures-dying-star-s-final-performance-in-fine-detail [2][3][4] is also interesting

To go back to image-to-image translation, you would not build a GAN to go from Hubble to JWST. At best some astronomers could be bluffed, but content-wise it would be worse than useless as the network would "invent" content that does not exists.

Going back to medical applications, I feel the same about most image to image applications. Can you clarify your thought on the topic?

---
[1] https://www.jwst.nasa.gov/content/about/comparisonWebbVsHubble.html
[2] https://en.wikipedia.org/wiki/Hubble_Space_Telescope#/media/File:NASA-HubbleSpaceTelescope-DeepField-2017.jpg
[3] https://en.wikipedia.org/wiki/Hubble_Space_Telescope#/media/File:Webb's_First_Deep_Field_(adjusted).jpg
[4] https://www.nasa.gov/image-feature/goddard/2022/nasa-s-webb-delivers-deepest-infrared-image-of-universe-yet

---
Misc:
- Can you confirm that all methods were implemented in a single, unified framework? Making it easy for re-use. Feel free to use tools such as https://anonymous.4open.science/ to share your code during rebuttal.
- Is the system for inflammation scoring, in Section 4, "hand-crafted" or "trainable". If the later, I assume it is only trained on real images, and not fine-tuned on synthetic images?
- Still in Section 4, I see that even the non-translated image gets a MSE of 0.031 (I think this is low?). So I am asking, is the selected task (of inflammation classification) a good benchmark and indication of the usefuleness of the "virtual" staining?
- In the sense, would performances crumble with virtual stains, when tested with a difficult application?
- Are you confindent that your findings would carry on other histopathology applications?

---
**Final rating:** I thank the authors for their response and clarification, and also reviewer `QQ4A` for the interactions (though the rebuttal ended-up before I could show-up again. While I keep my `strong acccept`, I am increasing my confidence from `3` to `4`. I very much look forward to continue the discussion at MIDL proper (though it has a 100% chance to deanonymize me, but hey, conferences are meant to share ideas).

---

### Meta-Review · Area_Chair_aq3G · 2023-02-24

**Recommendation:** Accept (Poster)
**Confidence:** 5

**Metareview:**

The authors propose a comprehensive evaluation of 12 different stain transfer methods for pixel-to-pixel translation in histopathology. The authors well-performed this comparison. I think that such comparative study is important for the field.